# Dental Phenotype with Minor Ectodermal Symptoms Suggestive of *WNT10A* Deficiency

**DOI:** 10.3390/children10020356

**Published:** 2023-02-10

**Authors:** Victoria-Eugenia García-Martínez, Ximo Galiana-Vallés, Otilia Zomeño-Alcalá, Raquel Rodríguez-López, Carmen Llena, María del Carmen Martínez-Romero, Encarna Guillén-Navarro

**Affiliations:** 1Alaquas Health Center, Departament General University Hospital, 46070 Valencia, Spain; 2Laboratory of Molecular Genetics, Clinical Analysis Service, Consortium General University Hospital, 46014 Valencia, Spain; 3Primary Care Dentistry, Departament General University Hospital, 46070 Valencia, Spain; 4Departament of Stomatology, Universitat de Valencia, 46010 Valencia, Spain; 5Molecular Genetics Section, Biochemistry and Clinical Genetics Center, University Clinical Hospital Virgen de la Arrixaca, Health Sciences PhD Program-UCAM, 30109 Murcia, Spain; 6IMIB-Pascual Parrilla, 30007 Murcia, Spain; 7CIBERER-ISCIII, 28029 Madrid, Spain; 8Faculty of Medicine and Health Sciences, UCAM Catholic University of Murcia, 30109 Murcia, Spain; 9Medical Genetics Section, Pediatrics Department, University Clinical Hospital Virgen de la Arrixaca, University of Murcia (UMU), 30120 Murcia, Spain

**Keywords:** ectodermal dysplasia, oligodontia, *WNT10A*, *EDAR*

## Abstract

Ectodermal dysplasias (EDs) represent a heterogeneous group of genetic disorders characterized by the abnormal development of ectodermal-derived tissues. They include the involvement of the hair, nails, skin, sweat glands, and teeth. Pathogenic variants in *EDA1* (Xq12–13.1; OMIM*300451), *EDAR* (2q11-q13; OMIM*604095), *EDARADD* (1q42-q43, OMIM*606603), and *WNT10A* (2q35; OMIM*606268) genes are responsible for most EDs. Bi-allelic pathogenic variants of *WNT10A* have been associated with autosomal recessive forms of ED, as well as non-syndromic tooth agenesis (NSTA). The potential phenotypic impact of associated modifier mutations in other ectodysplasin pathway genes has also been pointed out. We present on an 11-year-old Chinese boy with oligodontia, with conical-shaped teeth as the main phenotype, and other very mild ED signs. The genetic study identified the pathogenic variants *WNT10A* (NM_025216.3): c.310C > T; p. (Arg104Cys) and c.742C > T; p. (Arg248Ter) in compound heterozygosis, confirmed by parental segregation. In addition, the patient had the polymorphism *EDAR* (NM_022336.4): c.1109T > C, p. (Val370Ala) in homozygosis, named EDAR370. A prominent dental phenotype with minor ectodermal symptoms is very suggestive of *WNT10A* mutations. In this case, the EDAR370A allele might also attenuate the severity of other ED signs.

## 1. Introduction

The ectoderm is one of the three germ layers of the embryo. Around the third week of development, it differentiates to form the central and peripheral nervous system, skin, oral mucosa, tooth enamel, mucosa of the nostrils, sweat glands, hair, and nails, among other structures.

The specific differentiation of cells of ectodermal origin is regulated by very specific signaling pathways, such as WNT, BMP “bone morphogenic protein”, and FGF “fibroblast growth factor” pathways [1]. Structures of ectodermal origin (e.g., hair, teeth, nails) arise from cross-interactions between the ectodermal epithelium and the mesenchyme [2].

Ectodermal dysplasias (EDs) represent a heterogeneous group of genetic disorders characterized by the abnormal development of ectodermal-derived tissues, although most of them are also associated with abnormal development of mesoderm-derived structures and, sometimes, intellectual disability [3]. They are considered rare diseases, with a prevalence of 1:10,000 to 1:100,000. They can show any of the possible Mendelian inheritance patterns, and although clinical features are common to many of them, some syndromes have specific clinical findings. At present, about 100 separate EDs have been described [3]. Pathogenic variants in the *EDA1*, *EDAR*, *EDARADD,* and *WNT10A* genes are responsible for the majority of EDs [4].

Based on current genetic knowledge, it is possible to approach these rare pathologies from a molecular perspective [5]. The first genetic alteration identified as a cause of ED was the loss of the *EDA* gene [6]. Subsequent studies identified the EDA receptor defect (*EDAR*), the adaptor protein *EDARADD* “EDAR-associated death domain”, and *TRAF6* “TNF receptor-associated actor 6˝ genes [7,8].

The genetic basis of almost 50% of the conditions historically classified as EDs and the underlying causative genetic alterations in most of the most prevalent ED conditions are now known. In addition, it is now clear that many of the genes are affected in ED functions in common molecular pathways in the development of ectodermal derivatives. The categorization of EDs is complex, and different classification systems have succeeded each other by combining clinical and genetic data [3,9,10,11,12,13]. In the present proposed classification system by Wright, conditions are grouped based on the molecular pathway, the genotype, and the phenotype. The main groups are related to the EDA/NF-KappaB pathway, the WNT (*wingless-type)* pathway, the TP63 (*tumor protein p63*) pathway, and structural proteins. The rest of the EDs are included in a full list of almost 100 different conditions, which will require additional changes in the future due to the identification of new genes.

We present the clinical case of a boy in whom the first finding was the presence of conical teeth and oligodontia. These signs led to the diagnosis of mild ED, which was associated with *WNT10A* pathogenic variants and *EDAR* polymorphism.

## 2. Clinical Case

All procedures performed in this study were in accordance with the ethical standards of the institutional and/or national research committee and with the 1964 Helsinki Declaration and its later amendments or comparable ethical standards. Written informed consent was obtained from the parents for publication and for the presentation of clinical and radiographical images (Figure 1, Figure 2, Figure 3 and Figure 4).

We present the case of an 11-year-old Chinese boy who was referred to the dentist at the age of 23 months. In the first visit, it was noted that the boy was missing the left maxillary lateral incisor (6.2) and the maxillary and mandibular second molars (5.5, 6.5, 7.5, 8.5) and that he had conical-shaped anterior teeth (5.2, 7.2, 7.1, 8.1, 8.2) (Figure 1). The clinical examination results of his hair, eyes, eyebrows, nails, fingers, and skin were normal, except for dryness of the skin. His weight and height were at the 10th percentile. According to the parents, dental eruption was age-appropriate, at around 6 months of age. Psychomotor development was normal, and there were no previous episodes of fever or other significant pathologies. He was born at term after an uncomplicated pregnancy. The parents were nonconsanguineous. The paternal grandparents came from Daxue, China, and the maternal grandparents from Yuhu Village, China. The paternal grandparents came from Daxue, China, and the maternal grandparents from Yuhu Village, China.

In a new dental visit, at the age of 6 years, an orthopantomography was performed, where multiple agenesis was observed in both the primary and permanent dentition. Given that more than six permanent teeth were missing, it was labelled as oligodontia [14] (Figure 2). He also had a comprehensive phenotypic evaluation, and a genetic study was performed.

The child was followed up by the dentist. At present, the patient is 11 years old with normal psychomotor development, and his dental phenotype showed two conical maxillary central incisors and the first permanent molars. There were still conical-shaped anterior primary teeth and first primary molars. The germs of the second permanent molars were visible (Figure 3). The patient was then monitored by the dentist for treatment planning to address the growth and development of the jaws and subsequent rehabilitative treatment.

### 2.1. Phenotype Study

The phenotype was described using Human Phenotype Ontology (HPO) terms [15], which provide a standardized vocabulary of phenotypic abnormalities encountered in human disease. Abnormality of the primary teeth (HP:0006481) which had a conical-shaped and abnormality of the primary molar morphology (HP:0006344) that had an irregular coronal morphology and highly divergent roots were observed. Oligodontia (HP:0000677), tooth agenesis (HP:0009804), dental malocclusion (HP:0000689), abnormality of dental morphology (HP:0006482), the agenesis of mandibular premolars (HP:0011053), and a smooth tongue (HP:0010298) were also present. Abnormality of the skin (HP:0000951) was noted, being slightly atrophic with the loss of fingerprints on the thumbs of the hands. Mild keratosis pilaris (HP:0032152) was found on the cheeks and multiple hyperpigmented lentiginous macules <5 mm on the upper back and the buttocks. Likewise, erythema (HP:0010783) and some fissures were observed in the soles of the feet, suggestive of atopic pulpitis. The scalp hair was apparently normal, whose microscopic study showed a normal morphology and preserved birefringence. Sparse eyebrows (HP:0045075), palmoplantar keratoderma (HP:0000982) (Figure 4), and facial telangiectases (HP:0007380) were noted. No alterations in sweating, including hypohidrosis or hyperhidrosis, were present. See the HPO phenotypic description of the child in Table 1.

In the family dermatological evaluation, keratosis pilaris (HP:0032152) was observed in the father; plantar hyperkeratosis (HP:0007556) was observed in the mother; and dry skin (HP:0000958) was observed in the father, mother, and youngest son. To rule out possible anomalies in the size or number of teeth, the parents were comprehensively evaluated clinically and radiologically by a dentist. No significant alterations were found. 

### 2.2. Genetic Study

The methodology used was the massive parallel sequencing (next-generation sequencing or NGS) of all the coding and splicing regions of a total of 96 genes involved in the different types of ED. The test was performed by capture enrichment with specific probes (SureSelect XT^®^ Agilent) and subsequent sequencing in Illumina equipment (Miseq). Bioinformatic analysis was performed using Illumina Studio 3.0 Database Software to annotate variants: db SNP, 1000 genomes, ExAC, and Variant Server. Reference assembly (CRCh37/hg19). The minimum depth of filtered coverage in this analysis was 100X.

The genes included in the panel were: AXIN2, BRAF, CDH3, COL11A1, CTSC, CTSK, CYLN2, DKC1, DLX3, DSP, ED1, EDAR, EDAR2, EDARADD, EEC1 (ECE1), ELN, RCC2, ERCC3, EVC, EVC2, FGFR10, FGFR2, FGFR3, FLNA, GATA3, GAJ1, GJB2, GJB6, GTF2I, GTF2IRD1, GTF2IRD2, HRAS, IFT122, IFT43, INSR, KCTD1, KRAS2, KREMEN1, KRT14, KRT16, KRT17, KRT6A, KRT6B, KRTHB1, KRTHB3, KRTHB5, KRTHB6, LIMK1, LRP6, MBTS2, MEK1 = MAP2K1, MEK2 = MAP2K2, MSX1, NEMO = IKBKG, IKK1, IKK2, NFKB1, NFKB2, NOLA3 = NOP10, OFD1, PAX9, PIGL, PKP1, POC1A, PORCN, PVRL1, PVRL4, RECQL4, RFC2, RIPK4, RMRP, ROGDI, SETBP1, SHH, TBX3, TERC, TERT, TGF2H5, TINF2, TP63, TRAF6, TRPS1, TTDN1, TWIST2, UBR1, WDR19, WDR35, WHN y, and WNT10A.

Compound heterozygous variants, confirmed by parental segregation, were found in the *WNT10A* (NM_025216.2): c.310C > T; p. (Arg104Cys) (Clin Var: 532827, dbSNP: 764658964) *(ƒ = 0.0000517, gnomAD exomes v. 2.1.1)* of maternal origin and the c.742C > T; p. (Arg248Ter) (ClinVar: 265293; dbSNP: rs886039453) *(ƒ = 0.00000843, gnomAD Exomes v. 2.1.1)* of paternal origin in the patient’s germline DNA, in exons 2 and 3, respectively.

This result was confirmed by Sanger sequencing. Both variants were registered as pathogenic in public databases such as the Human Gene Mutation Database (HGMD^®^). In addition, the patient carried the *EDAR* (NM_022336.4): c.1109T > C; p. (Val370Ala) (ClinVar: 5858, dbSNP: rs3827760) *(ƒ = 0.154, gnomAD Exomes v. 2.1.1)* benign variant in homozygosis, as did both parents [16].

## 3. Discussion

Dental agenesis is one of the most common craniofacial anomalies. Depending on the number of missing teeth, it is considered hypodontia when less than six teeth are missing (excluding the third molars), oligodontia when more than six teeth are missing, or anodontia when all the teeth are missing [14]. Dental anomalies may be isolated [17] or syndromic. They can also be familial or occur sporadically.

According to the literature, pathogenic variants in the *WNT101A* gene lead to a wide clinical spectrum of ectodermal disorders. This wide genetic allelic heterogeneity involves at least three *WNT101A-related* phenotypes: odonto–onycho–dermal dysplasia (OODD, OMIM#257980), autosomal recessive (AR), Schöpf–Schulz–Passarge syndrome (SSPS, OMIM#224750), autosomal recessive AR and non-syndromic tooth agenesis (NSTA), or selective tooth agenesis type 4, (STHAG4, OMIM#150400) with an autosomal recessive or autosomal dominant hereditary pattern. One of the syndromic forms in which oligodontia or anodontia occurs is ED, which is also associated with dry skin, fine hair, and sweating problems. The most commonly associated ED genes are *WNT10A* [18] (Wnt family member 10 A), *EDA* (ectodysplasin A), *EDAR* (ectodysplasin A receptor) [19], and *EDARADD* (EDAR-associated death domain) [20,21]), which are also candidate genes for non-syndromic tooth agenesis (NSTA). In 2017 and 2019, keratinocyte differentiation factor l (*KDF1*) was also shown to result in ED [22,23]. Patients with mutations in *KDF1* present with abnormal skin, nails, and hair; a complete absence of permanent dental germs; and other abnormal ectodermal-derived tissues and organs.

Our patient presented with a prominent dental phenotype and minor ectodermal signs (mild skin anomalies, sparse eyebrows, and brittle nails), which were only identified under comprehensive dermatological evaluation. This phenotype was associated with pathogenic variants previously described, p. (Arg104Cys) [4] of maternal origin and p. (Arg248Ter) [24] of paternal origin observed in compound heterozygosis in the *WNT10A* gene. The frequency of the pathogenic variant inherited from the father, *WNT10A*: c.742C > T; p. (Arg248Ter), is precisely described in East Asian populations, with an overall frequency of 0.00084%.

The father comes from a region with a high potential for geographic isolation in eastern China, Daxue, a mountainous area of Tibet. The maternally inherited pathogenic variant, *WNT10A*: c.310C > T; p. (Arg104Cys), generates a less impactful amino acid change in the protein it encodes; its validated frequency is 0.065% in East Asian populations and it has not been identified in South Asia.

Both parents, as carriers, confirm the recessive inheritance pattern attributed in this case to the *WNT10A* gene. Neither of them presented with oligodontia or a complete ED phenotype, except dry skin (HP:0000958), keratosis pilaris (HP:0032152) in the father, and plantar hyperkeratosis (HP:0007556) in the mother. It should be noted that the paternal *WNT10A* pathogenic variant p. (Arg248Ter) is a nonsense mutation that leads to the end of protein synthesis, resulting in a smaller nonfunctional protein.

Pathogenic variants in the *WNT10A* gene have been associated with variable phenotypes, ranging from asymptomatic to a severe ED phenotype. Heterozygous pathogenic variants can lead to tooth agenesis. Homozygous or compound heterozygous *WNT10A* variants, as in the case of our patient, may lead to a wide phenotypic spectrum, from STHAG4 to odonto–onycho–dermal dysplasia and Schöpf–Schulz–Passarge syndrome [25]. The latter is a rare autosomal recessive ED characterized by palmoplantar keratoderma, hypotrichosis, hypodontia, nail dystrophy, and multiple apocrine hydrocystomas in the eyelids that develop with age [26], as well as adnexal skin tumors [27].

It has been observed that the number of missing teeth in the permanent dentition strongly depends on whether the affected individual is a homozygous or heterozygous carrier of mutations in the *WNT10A* gene. It is very likely that the phenotype depends on the characteristics and location of the genetic changes, and, consequently, on the deficiency generated in the *Wnt10* protein. In general, patients carrying biallelic nonsense pathogenic variants in *WNT10A* have a much more severe dental agenesis, whereas heterozygous individuals carrying a nonsense or a missense pathogenic variant are often unaffected or have a mild phenotype. Heterozygous compound patients were missing up to 6 permanent teeth, whereas homozygotes were usually missing 6 to 26 teeth, mostly around 16 [28]. This is consistent with our case, where the patient carried compound heterozygous variants in the *WNT10A* gene and showed severe dental involvement, while the parents did not present any dental involvement. Recently, *WNT10A*-linked oligo/hypodontia phenotypes have been described to be related with minor ectodermal manifestations, such as mild hair and nail anomalies, as described in our patient [29].

According to the literature, the compound heterozygous genotype in the *WNT10A* gene and its resulting phenotype that we report have not been previously described. Our patient could share a clinical condition with STHAG4 or odonto–onico–dermal dysplasia. A broad clinical spectrum has been defined among homozygous carriers of these variants separately, as a *WNT10A* homozygote genotype for c.310C > T; p. (Arg104Cys) delineated in ED-affected Turkish children [30] or an adult patient of Asian origin affected with SSPS [31]. Similarly, the entire clinical range has been observed in patients with the *WNT10A:* c. 742C > T; p. (Arg248Ter) homozygous variant associated with NSTA [32], OODD [33], or SSPS in an elderly patient [34]. Potentially, our patient could even evolve to this last condition. On the other hand, it has also been published that heterozygous carriers of these variants do not show dental loss, as it has also been observed in the parents of this patient [24,35]. Table 2 presents the clinical features described in cases with referred variants in homozygous or compound heterozygous.

Regarding the variant in the *EDAR* gene: c. 1109T > C; p. (Val370Ala), it has been suggested that it could act as a modifying variant of the ED phenotype in patients carrying causal variants in the *EDA* gene, one of the main susceptibility genes for ED [36]. It is difficult to attribute the very mild ectodermal manifestations in our patient to the protective effect of the homozygous EDAR370A allele, since there are not specific analyzed cohorts to establish such an association with *WNT10A* gene mutation carriers. However, we cannot exclude it, given the relationship among the different molecular pathways involved in ectodermal derivatives development. It is difficult to attribute to the fact that the patient is a carrier of such a homozygous polymorphism, the few symptoms that he presents at the dermatological level or in other structures, since there are no series analyzed to establish such an association with mutation carriers in the *WNT10A* gene.

According to the functional prediction software for genetic variants Alamut Software, version 1.5.1, the physical–chemical characteristics of the T/C nucleotide change that generates the polymorphism could be important for the *Edar* protein that it encodes.

The fact that both parents were homozygous for the EDAR370A allele, and therefore the patient, is not surprising. The frequency of this variant in the population from which the patient’s ancestors came from has reversed to become the consensus allele, with 92.1% described in East Asian populations. The allelic frequencies described for the C allele in European, African, and Caucasian populations have remained between 1 and 2%. Its genetic drift shows great contrasts between different populations and ethnic groups. Some authors have suggested that its homozygous genotype could confer some advantage. Its evolutionary conservation is evident in the population of origin of the patient’s family [37].

## 4. Conclusions

Our patient presented a prominent dental phenotype and minor ectodermal signs carrying biallelic *WNT10A* pathogenic variants in compound heterozygosis. This supports previous reports and highlights the importance of searching skin and nail anomalies in patients consulting for apparently isolated dental anomalies to establish an accurate diagnosis during childhood. The association of the EDAR370A homozygous allele might also attenuate the severity of ED signs in this patient, although more related data are needed.

## Figures and Tables

**Figure 1 children-10-00356-f001:**
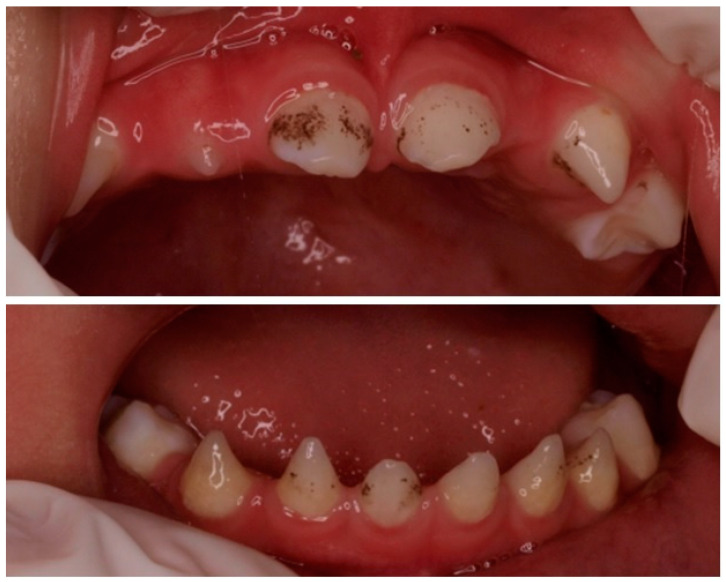
Presence of conical anterior temporary teeth. Absence of the left lateral maxillary incisor and the maxillary and mandibular second molars.

**Figure 2 children-10-00356-f002:**
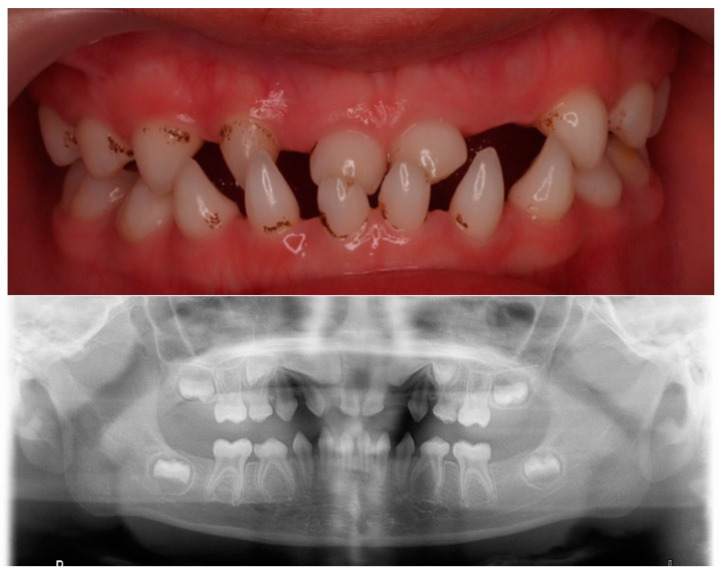
Clinical and radiographical findings at 6 years old.

**Figure 3 children-10-00356-f003:**
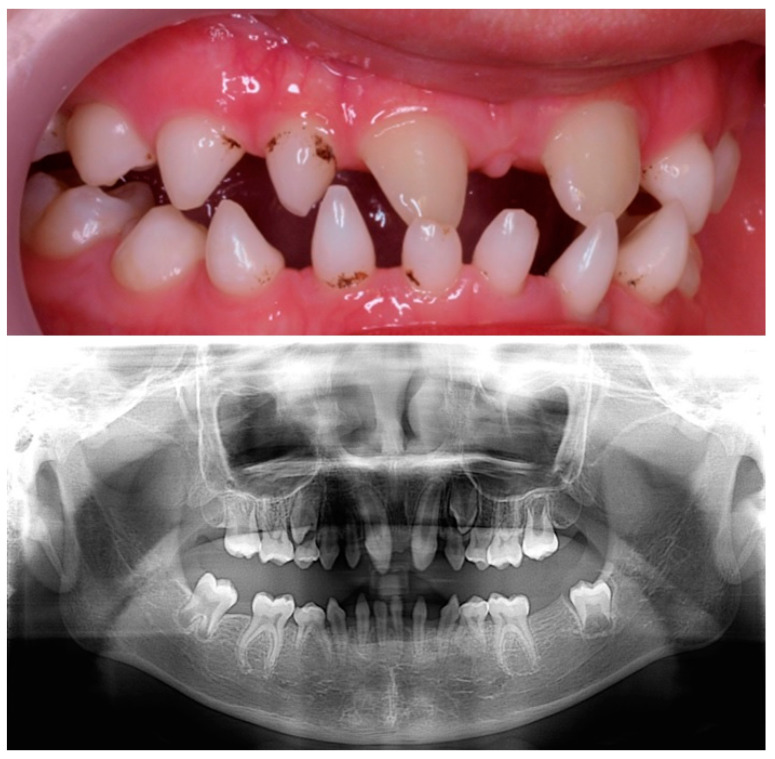
Clinical and radiographical findings at 11 years old.

**Figure 4 children-10-00356-f004:**
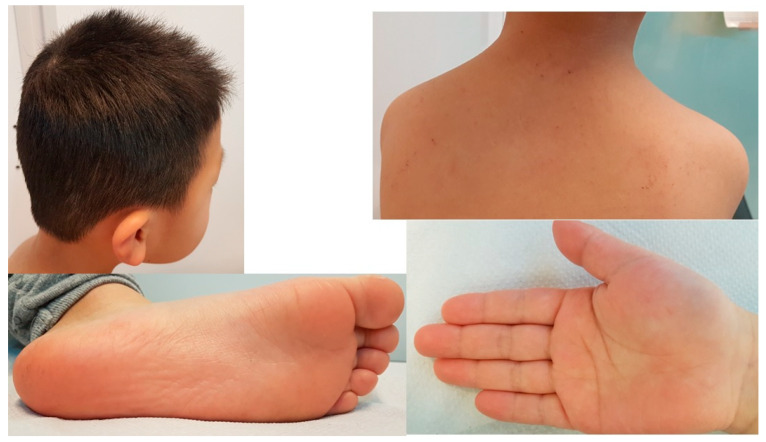
Erythema and some fissures on the soles of the feet suggestive of atopic pulpitis.

**Table 1 children-10-00356-t001:** Comprehensive phenotypic description of the patient based on the *WNT10A* gene using Human Phenotype Ontology (HPO).

HPO_TERM_ID	HPO_TERM_NAME	Patient	Description
HP:0006344	Abnormality of primary molar morphology	YES	Irregular coronal morphology and highly divergent roots.
HP:0006481	Abnormality of primary teeth	YES	Front teeth are conical-shaped.
HP:0011056	Agenesis of first permanent molar tooth	NO	
HP:0001798	Anonychia	NO	
HP:0006482	Abnormality of dental morphology	YES	
HP:0001792	Small nails	NO	
HP:0002209	Sparse scalp hair	NO	
HP:0000478	Abnormality of the eye	NO	
HP:0010298	Smooth tongue	YES	
HP:0011053	Agenesis of mandibular premolar	YES	Also agenesis of the maxillary premolars.
HP:0008388	Abnormal toenail morphology	NO	
HP:0000668	Hypodontia	NO	
HP:0001806	Onycholysis	NO	
HP:0031405	Poroma	NO	
HP:0000975	Hyperhidrosis	NO	
HP:0002671	Basal cell carcinoma	NO	
HP:0008391	Dystrophic fingernails	NO	
HP:0005216	Impaired mastication	NO	
HP:0000007	Autosomal recessive inheritance		
HP:0000613	Photophobia	NO	
HP:0001231	Abnormal fingernail morphology	NO	
HP:0010764	Short eyelashes	NO	
HP:0006342	Peg-shaped maxillary lateral incisors	YES	
HP:0006289	Agenesis of central incisor	NO	
HP:0009804	Tooth agenesis	YES	1.4,1.5, 2.2, 2.4, 2.5, 3.1, 3.2, 3.3, 3.4, 3.5, 4.1, 4.2, 4.3, 4.4, 4.5, 5.5, 6.2, 6.5,7.5, 8.5.
HP:0001596	Alopecia	NO	
HP:0001810	Dystrophic toenails	NO	
HP:0000951	Abnormality of the skin	YES	Multiple hyperpigmented lentiginous macules <5 mm on the upper back and buttocks. Appearance was slightly atrophic with loss of fingerprints on the thumbs of the hands.
HP:0031454	Apocrine hidrocystoma	NO	
HP:0000684	Delayed eruption of teeth	NO	
HP:0000320	Bird-like facies	NO	
HP:0002231	Sparse body hair	NO	
HP:0011219	Short face	NO	
HP:0000202	Oral cleft	NO	
HP:0000685	Hypoplasia of teeth	NO	
HP:0000958	Dry skin	YES	
HP:0002860	Squamous cell carcinoma	NO	
HP:0012472	Eclabion	NO	
HP:0000968	Ectodermal dysplasia	YES	
HP:0006297	Enamel hypoplasia	NO	
HP:0002164	Nail dysplasia	YES	Brittle toenails.
HP:0001595	Abnormal hair morphology	NO	
HP:0011313	Narrow nails	NO	
HP:0008070	Sparse hair	NO	
HP:0001807	Ridged nails	NO	
HP:0006323	Premature loss of primary teeth	NO	
HP:0007380	Facial telangiectasia	NO	
HP:0000677	Oligodontia	YES	
HP:0000689	Dental malocclusion	YES	
HP:0000691	Microdontia	NO	
HP:0000696	Delayed eruption of permanent teeth	NO	
HP:0100840	Aplasia/hypoplasia of the eyebrow	NO	
HP:0002213	Fine hair	NO	
HP:0007410	Palmoplantar hyperhidrosis	NO	
HP:0010783	Erythema	YES	Erythema and some fissures were observed in the balls of the feet, suggestive of atopic pulpitis.
HP:0007556	Plantar hyperkeratosis	NO	
HP:0011359	Dry hair	NO	
HP:0000679	Taurodontia	NO	
HP:0000687	Widely spaced teeth	YES	
HP:0000690	Agenesis of maxillary lateral incisors	YES	Agenesis of a maxillary lateral incisor, temporary and permanent.
HP:0032152	Keratosis pilaris	YES	Mild keratosis pilaris on the cheeks.
HP:0001816	Thin nails	NO	
HP:0025114	Hypergranulosis	NO	
HP:0000982	Palmoplantar keratoderma	NO	
HP:0025092	Epidermal acanthosis	NO	
HP:0006336	Short dental root	NO	
HP:0100615	Ovarian neoplasm	NO	
HP:0040162	Orthokeratosis	NO	
HP:0045075	Sparse eyebrow	YES	Sparse hair in eyebrow tail.
HP:0000966	Hypohidrosis	NO	
HP:0006349	Agenesis of permanent teeth	YES	
HP:0011051	Agenesis of premolar	YES	
HP:0011078	Abnormality of canine	YES	

**Table 2 children-10-00356-t002:** Clinical features described in cases with referred variants in homozygous or compound heterozygous.

Cases Reported		Güven et al., 2019 [29]	Hsu et al., 2018 [30]	Yang et al., 2015 [34]	Yu et al., 2019 [32]	Zimmermann et al., 2017 [33]	Novel Case
Sex		F	M	M	F	M	M
Age at diagnosis (years)		DNA (child)	54	8.5	14	53	11
Tooth agenesis							
HP:0006482	Abnormality of dental morphology	YES	YES	YES	YES	YES	YES
HP:0006349	Agenesis of permanent teeth	YES	YES	YES	YES	YES	YES
HP:0000677	Oligodontia	YES	YES	YES	YES	YES	YES
Sweating							
HP:0000966	Hipohidrosis	NO	NO	NO	YES	NO	NO
HP:0007410	Palmoplantar hyperhidrosis	YES	NO	NO	NO	YES	NO
Skin							
HP:0000958	Dry skin	YES	YES	NO	YES	YES	YES
HP:0000982	Palmoplantar keratoderma	NO	YES	NO	NO	YES	NO
HP:0031454	Apocrine hidrocystoma	NO	YES	NO	NO	YES	NO
Hair							
HP:0002209	Sparse scalp hair	YES	YES	NO	NO	YES	NO
HP:0002231	Sparse body hair	ND	YES	NO	NO	YES	YES
Nails							
HP:0002164	Nail dysplasia	YES	YES	NO	YES	YES	YES
Clinical Diagnosis		ED	SSPS	STHAG	OODD	SSPS	STHAG with mild ED
Variants in *WNT10A* (NM_025216.3)		c. 310 C > A	c. 310 C > A	c. 310 C > A/c. 637 T > A	c. 742 C > T	c. 742 C > T/c. 321 C > A	c. 310 C > A/c. 742 C > T
Protein change		*p. (Arg104Cys)*	*p. (Arg104Cys)*	*p. (Arg104Cys)/p. (Gly213Ser)*	*p. (Arg248Ter)*	*p. (Arg248Ter)/p. (Cys107Ter)*	*p. (Arg104Cys)/p. (Arg248Ter)*
Zigosity		*Homozygote*	*Homozygote*	*Compound heterozygous*	*Homozygote*	*Compound heterozygous*	*Compound heterozygous*

F, female; M, male; DNA, data not available; ED, ectodermal dysplasia; SSPS, Schöpf–Schulz–Passarge syndrome; STHAG, selective tooth agenesis type 4; OODD, odonto–onycho–dermal dysplasia.

## Data Availability

The data provided are clinical and analytical data derived from the genetic study, which are described in the body of the document. There are no other data to be provided.

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
