# Peer review of "Dental Phenotype with Minor Ectodermal Symptoms Suggestive of WNT10A Deficiency"

_children, 2023, doi:10.3390/children10020356_

Round 1

Reviewer 1 Report

In this manuscript, García-Martínez et al., report a Chinese 11-year-old boy with mainly oligodontia and conical-shaped teeth phenotype related to a compound heterozygous WNT10A1 genotype confirmed by parental analysis, and involving two previously reported pathogenic variants: NM_025216.3(WNT10A):c.[310C>T];[742C>T] or p.[Arg104Cys];[Arg248Ter]. This genotype was documented by massively parallel sequencing or NGS of all the coding and splicing regions of a total of 96 genes involved in ectodermal dysplasias. The resulting patient´s dental and dermatological phenotype is well described and illustrated.

I have the following comments:

- Please cite all OMIM ID entries for each locus (OMIM*) and their related disorders (OMIM#) when they are mentioned for the first time, even in abstract.

- Please include the reference sequence employed for variant annotation (NM_025216.3?)

- Please include in the results, the allelic frequencies obtained at NGS for each WNT10A1 variants.

- According to the literature, please emphasize if the herein identified compound heterozygous WNT10A1 genotype, and its resulting phenotype, has bee,n or not, previously described. This aspect supports the novelty of your clinical report.

- Pathogenic variants in the WNT101A gene lead to a wide clinical spectrum of ectodermal disorders. This wide genetic allelic heterogeneity, involves at least three WNT101A-related phenotypes (https://www.omim.org/clinicalSynopsis/table?mimNumber=257980,224750,150400):

a) ODONTOONYCHODERMAL DYSPLASIA (OODD, OMIM#257980). Autosomal recessive.

b) SCHOPF-SCHULZ-PASSARGE SYNDROME (SSPS, OMIM#224750). Autosomal recessive.

c) TOOTH AGENESIS, SELECTIVE, type 4 (STHAG4, OMIM#150400). Autosomal recessive or autosomal dominant.

Particularly, STHAG4 (OMIM#150400) disease, shares the clinical picture identified in the present patient (dental abnormalities with minimal dermatological involvement). Thus, the authors must discuss the possibility of a differential diagnosis, as your patient highly resembles the STHAG4 clinical picture.

This aspect may be helped by including a comparative Table with the previously described clinical features for STHAG4 (including percentages) and the observed clinical charachteristics of his patient.

- Please cite directly in the text, at their first mention, the ClinVar and dnSNP entries for both identified WNT10A1 variants (i.e. Arg104Cys-ClinVar: 532827; dbSNP: rs764658964; and Arg248*-ClinVar:265293; dbSNP:rs886039453), even in abstract to proper/better further citation and indexing. Thus, cites for both variants mentioned in these genotypic databases could be deleted at reference section.

- The documented oligodontia with very mild dermatological phenotype has been previously linked to NM_025216.3(WNT10A):c.310C>T or p.(Arg104Cys) variant. This variant needs further discussion, as the phenotype resulting from a homozygous p.(Arg104Cys) WNT10A1 genotype, has been previously delineated in an adult Asian-derived patient.

Thus, please cite and discuss the findings of:

Hsu TC, Lee JY, Hsu MM, Chao SC. Case report of Schöpf-Schulz-Passarge syndrome resulting from a missense mutation, p.Arg104Cys, in WNT10A. J Dermatol. 2018 Apr;45(4):475-478. doi: 10.1111/1346-8138.14201. Epub 2017 Dec 22. PMID: 29271000.

- As heterozygous WNT10A1 individuals, could have mild dental manifestations, please emphasized in the material and methods section, if both parents were subjected to comprehensive dentistry evaluation, including orthopantomography, as this aspect is only briefly mentioned in the discussion section. It could be interesting to discuss if heterozygous individuals for Arg104Cys or Arg248Ter WNT101A variants have been reported in literature with minimal dental or other dermatological features related to ectodermal dysplasias.

- Several texts in spanish languaje must be removed along text.

Author Response

In this manuscript, García-Martínez et al., report a Chinese 11-year-old boy with mainly oligodontia and conical-shaped teeth phenotype related to a compound heterozygous WNT10A1 genotype confirmed by parental analysis, and involving two previously reported pathogenic variants: NM_025216.3(WNT10A):c.[310C>T];[742C>T] or p.[Arg104Cys];[Arg248Ter]. This genotype was documented by massively parallel sequencing or NGS of all the coding and splicing regions of a total of 96 genes involved in ectodermal dysplasias. The resulting patient´s dental and dermatological phenotype is well described and illustrated.

Thank you for your comments and suggestions for improvement. Here, is a point-by-point response to your comments.

I have the following comments:

- Please cite all OMIM ID entries for each locus (OMIM*) and their related disorders (OMIM#) when they are mentioned for the first time, even in abstract.

All OMIN ID for each locus (OMIM*) and their related disorders, have been cited in the text.

- Please include the reference sequence employed for variant annotation (NM_025216.3?)

Reference sequences have been added in the text.

- Please include in the results, the allelic frequencies obtained at NGS for each WNT10A1 variants.

Allelic frequencies have been added in the results section.

- According to the literature, please emphasize if the herein identified compound heterozygous WNT10A1 genotype, and its resulting phenotype, has been or not, previously described. This aspect supports the novelty of your clinical report.

A paragraph has been added in the discussion section.

- Pathogenic variants in the WNT101A gene lead to a wide clinical spectrum of ectodermal disorders. This wide genetic allelic heterogeneity, involves at least three WNT101A-related phenotypes (https://www.omim.org/clinicalSynopsis/table?mimNumber=257980,224750,150400):

  1. a) ODONTOONYCHODERMAL DYSPLASIA (OODD, OMIM#257980). Autosomal recessive.
  2. b) SCHOPF-SCHULZ-PASSARGE SYNDROME (SSPS, OMIM#224750). Autosomal recessive.
  3. c) TOOTH AGENESIS, SELECTIVE, type 4 (STHAG4, OMIM#150400). Autosomal recessive or autosomal dominant.

Particularly, STHAG4 (OMIM#150400) disease, shares the clinical picture identified in the present patient (dental abnormalities with minimal dermatological involvement). Thus, the authors must discuss the possibility of a differential diagnosis, as your patient highly resembles the STHAG4 clinical picture.

A new paragraph in the discussion section with new references have been added.

This aspect may be helped by including a comparative Table with the previously described clinical features for STHAG4 (including percentages) and the observed clinical charachteristics of his patient.

A comparative table with the clinical features described in cases with referred variants in homozygous or compound heterozygous has been added.

- Please cite directly in the text, at their first mention, the ClinVar and dnSNP entries for both identified WNT10A1 variants (i.e. Arg104Cys-ClinVar: 532827; dbSNP: rs764658964; and Arg248*-ClinVar:265293; dbSNP:rs886039453), even in abstract to proper/better further citation and indexing. Thus, cites for both variants mentioned in these genotypic databases could be deleted at reference section.

The ClinVar and dnSNP entries for both identified WNT10A1 variants have been added and  reference has been deleted.

- The documented oligodontia with very mild dermatological phenotype has been previously linked to NM_025216.3(WNT10A):c.310C>T or p.(Arg104Cys) variant. This variant needs further discussion, as the phenotype resulting from a homozygous p.(Arg104Cys) WNT10A1 genotype, has been previously delineated in an adult Asian-derived patient.

The paragraph has been completed in the discussion section.

-Thus, please cite and discuss the findings of:

Hsu TC, Lee JY, Hsu MM, Chao SC. Case report of Schöpf-Schulz-Passarge syndrome resulting from a missense mutation, p.Arg104Cys, in WNT10A. J Dermatol. 2018 Apr;45(4):475-478. doi: 10.1111/1346-8138.14201. Epub 2017 Dec 22. PMID: 29271000.

The reference has been included and discussed .

- As heterozygous WNT10A1 individuals, could have mild dental manifestations, please emphasized in the material and methods section, if both parents were subjected to comprehensive dentistry evaluation, including orthopantomography, as this aspect is only briefly mentioned in the discussion section.

A paragraph has been added in the phenotype section description related to the dental evaluation of the parents.

-It could be interesting to discuss if heterozygous individuals for Arg104Cys or Arg248Ter WNT101A variants have been reported in literature with minimal dental or other dermatological features related to ectodermal dysplasias.

Two references of healthy carriers are included 34 and 35.

- Several texts in spanish languaje must be removed along text.

The text in Spanish has been removed.

Reviewer 2 Report

Thank you for your paper “Analysis of oral health status in elite athletes. Differences between individual and team sports”. The article is interesting. Nevertheless, I suggest significant improvements. The article needs to be rewritten.

Line 6: the author’s affiliation is written as: “Valencia and Spain”, please write the coma instead of „and”

Lines 78-79: “….6.2 and the maxillary and mandibular second molars 78 and conical-shaped anterior teeth”

please specify which teeth: ….maxilary left lateral incisor (62), the maxillary and mandibular second molars (55, 65, 75, 85) and conical-shaped anterior teeth ( ……………..

Line 79-80: “The rest of the clinical examination at that time was referred as normal, except for dryness of the skin” – what kind of examinations were done?

Lines 88-95, 222-250 are in Spanish

Lines 127-132 should be in the place where the description of dental status is – it seems to be the repetition.

The Clinical case section should be rewritten. It takes work to read. Maybe it should be written as: the first dental visit was….., the second, or the kid was under control, the medical history,

I would suggest to the inclusion of the following work to increase the bibliographic:

doi: 10.5114/ada.2020.100480

Author Response

Thank you for your paper “Analysis of oral health status in elite athletes. Differences between individual and team sports”. The article is interesting. Nevertheless, I suggest significant improvements. The article needs to be rewritten.

Thank you for your comments. To the manuscript entitled: Dental phenotype with minor ectodermal symptoms suggestive of WNT10A deficiency. Here, is a point-by-point response to your comments.

Line 6: the author’s affiliation is written as: “Valencia and Spain”, please write the coma instead of „and”

“and” has been deleted and substituted by a coma.

Lines 78-79: “….6.2 and the maxillary and mandibular second molars and conical-shaped anterior teeth” please specify which teeth: ….maxillary left lateral incisor (62), the maxillary and mandibular second molars (55, 65, 75, 85) and conical-shaped anterior teeth ( ……………..

All teeth have been specified with their description-position and number accordingly FDI identification

Line 79-80: “The rest of the clinical examination at that time was referred as normal, except for dryness of the skin” – what kind of examinations were done?

Clinical examination of hair, eyes, eyebrows, nails, fingers and skin were normal, except for dryness of the skin

Lines 88-95, 222-250 are in Spanish

Lines have been deleted.

Lines 127-132 should be in the place where the description of dental status is – it seems to be the repetition.

A description about the dental status of the parents has been added accordingly with the reviewer 1 suggestion at the end of the first paragraph of the clinical case description

The Clinical case section should be rewritten. It takes work to read. Maybe it should be written as: the first dental visit was….., the second, or the kid was under control, the medical history,

The clinical case section has been reordered. In the first part it has been included the dental aspects of the child during their follow up, in the following subsections, the phenotypic and genotypic description of the child and parents is included. The other of the figures 3 and 4 have been changed.

I would suggest to the inclusion of the following work to increase the bibliographic: doi: 10.5114/ada.2020.100480.

We have not found any association between the article you recommend (Diagnosis of Papillon-Lefèvre syndrome: review of the literature and case report) and our case report, so we are sorry we cannot include it.

Round 2

Reviewer 1 Report

All criticism were fully and satisfactorily addressed.

Just please write the protein changes accordingly HGVS nomenclature (https://varnomen.hgvs.org/recommendations/protein/variant/substitution/), i.e. p.Arg104Cys, must be written as p. (Arg104Cys), and so on.

Please include the sign "#" to describe ID OMIM phenotypes, and "*" for responsible loci.

Please clarify gene referenced in Table 2, and in its foot table, change "no datos" by "data not available".

Lane 247: please describe correctly the incomplete description for the "c.1109>C" nucleotide change.

Preferably please specify the abbreviaton "EDAR370A" (lane 251) allele accordingly to HGVS nomenclature, when this it is described for fisrt time. 

Author Response

Just please write the protein changes accordingly HGVS nomenclature (https://varnomen.hgvs.org/recommendations/protein/variant/substitution/), i.e. p.Arg104Cys, must be written as p. (Arg104Cys), and so on.

R-Proteine changes has been writen accordingly your recomendation.

Please include the sign "#" to describe ID OMIM phenotypes, and "*" for responsible loci.

R-Sign # has been included to describe ID OMIM phenotype and * for responsible loci.

Please clarify gene referenced in Table 2, and in its foot table, change "no  datos" by "data not available".

R-Gene has been referenced and the foot note table has been changed

Lane 247: please describe correctly the incomplete description for the "c.1109>C" nucleotide change.

R-Description has been completed as c.1109T>C

Preferably please specify the abbreviaton "EDAR370A" (lane 251) allele accordingly to HGVS nomenclature, when this it is described for fisrt time. 

R-This alelle is complete described in the abstrac when this it is described for fisrt time.

Reviewer 2 Report

Thank you for the correction. 

Author Response

None